# Early versus deferred anti-SARS-CoV-2 convalescent plasma in patients admitted for COVID-19: A randomized phase II clinical trial

**María Elvira Balcells**[1]☯*, **Luis Rojas**[2,3]☯, **Nicole Le Corre**[4,5], **Constanza Martínez-Valdebenito**[4,5], **María Elena Ceballos**[1], **Marcela Ferrés**[4,5], **Mayling Chang**[6], **Cecilia Vizcaya**[4], **Sebastián Mondaca**[6], **Álvaro Huete**[7], **Ricardo Castro**[8], **Mauricio Sarmiento**[6], **Luis Villarroel**[9], **Alejandra Pizarro**[1], **Patricio Ross**[2], **Jaime Santander**[10], **Bárbara Lara**[11], **Marcela Ferrada**[12], **Sergio Vargas-Salas**[6], **Carolina Beltrán-Pavez**[13,14], **Ricardo Soto-Rifo**[13,14], **Fernando Valiente-Echeverría**[13,14], **Christian Caglevic**[15], **Mauricio Mahave**[15], **Carolina Selman**[15], **Raimundo Gazitúa**[15], **José Luis Briones**[15], **Franz Villarroel-Espindola**[15,16], **Carlos Balmaceda**[17], **Manuel A. Espinoza**[9], **Jaime Pereira**[6], **Bruno Nervi**[6]

1 Department of Infectious Diseases, School of Medicine, Pontificia Universidad Católica de Chile, Santiago, Chile, 2 Department of Internal Medicine, School of Medicine, Pontificia Universidad Católica de Chile, Santiago, Chile, 3 Program of Pharmacology and Toxicology, School of Medicine, Pontificia Universidad Católica de Chile, Santiago, Chile, 4 Department of Pediatric Infectious Diseases and Immunology, School of Medicine, Pontificia Universidad Católica de Chile, Santiago, Chile, 5 Diagnostic Virology Laboratory, Red de Salud UC CHRISTUS, Santiago, Chile, 6 Department of Hematology and Oncology, School of Medicine, Pontificia Universidad Católica de Chile, Santiago, Chile, 7 Department of Radiology, School of Medicine, Pontificia Universidad Católica de Chile, Santiago, Chile, 8 Department of Intensive Care Medicine, School of Medicine, Pontificia Universidad Católica de Chile, Santiago, Chile, 9 Department of Public Health, School of Medicine, Pontificia Universidad Católica de Chile, Santiago, Chile, 10 Department of Psychiatry, School of Medicine, Pontificia Universidad Católica de Chile, Santiago, Chile, 11 Emergency Medicine Section, School of Medicine, Pontificia Universidad Católica de Chile, Santiago, Chile, 12 Clinical Research Center, School of Medicine, Pontificia Universidad Católica de Chile, Santiago, Chile, 13 Laboratory of Molecular and Cellular Virology, Virology Program, Institute of Biomedical Sciences, Faculty of Medicine, Universidad de Chile, Santiago, Chile, 14 HIV/AIDS Work Group, Faculty of Medicine, Universidad de Chile, Santiago, Chile, 15 Instituto Oncológico Fundación Arturo López Pérez, Santiago, Chile, 16 Translational Medicine Research Laboratory, Fundación Arturo López Pérez, Santiago, Chile, 17 Health Technology Assessment Unit, Clinical Research Center, School of Medicine, Pontificia Universidad Católica de Chile, Santiago, Chile

☯ These authors contributed equally to this work.
* ebalcells@uc.cl

**Data Availability Statement:** We are unable to make the data available in a public repository or uploaded as supplementary information because

## Abstract

### Background

Convalescent plasma (CP), despite limited evidence on its efficacy, is being widely used as a compassionate therapy for hospitalized patients with COVID-19. We aimed to evaluate the efficacy and safety of early CP therapy in COVID-19 progression.

### Methods and findings

The study was an open-label, single-center randomized clinical trial performed in an academic medical center in Santiago, Chile, from May 10, 2020, to July 18, 2020, with final follow-up until August 17, 2020. The trial included patients hospitalized within the first 7 days of COVID-19 symptom onset, presenting risk factors for illness progression and not on mechanical ventilation. The intervention consisted of immediate CP (early plasma group)

this is not permitted by our organisation's research governance policy and Ethics committee regulations. Anonymised data can be made available to researchers who meet the conditions of the ethics approval and research governance policy that applies to this study. Researchers may request anonymized data access by contacting Universidad Católica's School of Medicine Research Office (DIDEMUC) at didemuc@med.puc.cl.

**Funding:** This work was supported by a grant from the Fondo de Adopción Tecnológica SiEmpre, SOFOFA Hub (https://web.sofofa.cl/centros-sofofa/sofofa-hub/), and Ministerio de Ciencia, Tecnología, Conocimiento e Innovación, Chile (https://www.minciencia.gob.cl/)(M.E.B) as well as a donation from ENEL Chile S.A. (www.enel.cl/es/sostenibilidad.html). The funders had no role in study design, data collection and analysis, decision to publish, or preparation of the manuscript.

**Competing interests:** I have read the journal's policy and the authors of this manuscript have the following competing interests: -S.M. is a consultant for Foundation Medicine and Roche; he also has received research funding from Bristol-Myers Squibb and Foundation Medicine. - C.C is the Head of Cancer Research Department at Instituto Oncológico Fundación Arturo López Pérez (FALP) and declares that FALP has received funding from "Confederación de la producción y el Comercio (CPC)" to develop research based on Convalescent Plasma from COVID-19 recovered patients to treat patients with active COVID-19 infection. None of the authors from FALP has received any payment for their participation in this publication nor for participating in the trial in their investigator roles.

**Abbreviations:** CP, convalescent plasma; CRP, C-reactive protein; ICU, intensive care unit; ID50, 50% inhibitory dose; IRR, incidence rate ratio; ITT, intention to treat; LDH, lactate dehydrogenase; NAb, neutralizing antibody; OD, optical density; OR, odds ratio; pro-BNP, pro-B type natriuretic peptide; SAE, serious adverse event; SOFA, Sequential Organ Failure Assessment; TRALI, transfusion-associated acute lung injury.

versus no CP unless developing prespecified criteria of deterioration (deferred plasma group). Additional standard treatment was allowed in both arms. The primary outcome was a composite of mechanical ventilation, hospitalization for >14 days, or death. The key secondary outcomes included time to respiratory failure, days of mechanical ventilation, hospital length of stay, mortality at 30 days, and SARS-CoV-2 real-time PCR clearance rate. Of 58 randomized patients (mean age, 65.8 years; 50% male), 57 (98.3%) completed the trial. A total of 13 (43.3%) participants from the deferred group received plasma based on clinical aggravation. We failed to find benefit in the primary outcome (32.1% versus 33.3%, odds ratio [OR] 0.95, 95% CI 0.32–2.84, $p > 0.999$) in the early versus deferred CP group. The in-hospital mortality rate was 17.9% versus 6.7% (OR 3.04, 95% CI 0.54–17.17 $p = 0.246$), mechanical ventilation 17.9% versus 6.7% (OR 3.04, 95% CI 0.54–17.17, $p = 0.246$), and prolonged hospitalization 21.4% versus 30.0% (OR 0.64, 95% CI, 0.19–2.10, $p = 0.554$) in the early versus deferred CP group, respectively. The viral clearance rate on day 3 (26% versus 8%, $p = 0.204$) and day 7 (38% versus 19%, $p = 0.374$) did not differ between groups. Two patients experienced serious adverse events within 6 hours after plasma transfusion. The main limitation of this study is the lack of statistical power to detect a smaller but clinically relevant therapeutic effect of CP, as well as not having confirmed neutralizing antibodies in donor before plasma infusion.

## Conclusions

In the present study, we failed to find evidence of benefit in mortality, length of hospitalization, or mechanical ventilation requirement by immediate addition of CP therapy in the early stages of COVID-19 compared to its use only in case of patient deterioration.

## Trial registration

NCT04375098.

---

## Author summary

### Why was this study done?

- The severe acute respiratory syndrome coronavirus 2 (SARS-CoV-2) pandemic has become a matter of worldwide concern, and except for corticosteroids, no other validated treatment against SARS-CoV-2 has been found so far.

- Plasma from convalescent patients containing antibodies against the virus is being widely used as a treatment alternative against this virus, but few randomized clinical trials have been carried out to show any clinical benefit for patients with COVID-19.

### What did the researchers do and find?

- We conducted a randomized clinical trial. Fifty-eight hospitalized patients in the early stages of COVID-19 ($\leq$7 days of symptoms) and with a high risk of progression into respiratory failure were recruited.

- Patients were randomized into 2 groups: The early plasma group received convalescent plasma at enrollment, and the deferred plasma group received convalescent plasma only in case of respiratory aggravation or if the patient still required hospitalization for symptomatic COVID-19 >7 days after enrollment.

- The proportion achieving the combined primary outcome of mechanical ventilation, prolonged hospitalization, or in-hospital death was 32.1% with immediate plasma versus 33.3% in the arm deferring plasma until aggravation, a non-significant difference.

## What do these findings mean?

- Our study failed to show that early convalescent plasma administration improves the outcome compared to convalescent plasma use only in case of clinical deterioration.

- The small sample size of the study precludes any definitive conclusions, but the results are in agreement with observations from other trials on convalescent plasma for patients hospitalized with COVID-19.

## Introduction

The SARS-CoV-2 pandemic resulted in over 24 million infections and 833,000 deaths by August 29, 2020 [1]. During the early months of the pandemic, case series and cohorts from China and the United States analyzed demographic and outcome data for hundreds of inpatients admitted for COVID-19. These showed an intensive care unit (ICU) admission rate between 5% and 26%, and overall mortality from 1.4% to 28.3% [2,3]. Older age, male sex, and preexisting hypertension and/or diabetes rapidly stood out among risk factors correlating with case fatality rate, in the first large case series of sequentially hospitalized patients with confirmed COVID-19 in the US [4]. The scientific community is desperate to find effective treatments and immunization against SARS-CoV-2, and so far, dexamethasone is the only drug that has shown a survival benefit, among those patients who are receiving either invasive mechanical ventilation or oxygen alone at randomization [5]. The antiviral remdesivir has shown a shorter time to recovery in adults hospitalized with COVID-19 and with evidence of lower respiratory tract infection, but its effect on overall mortality remains controversial [6]. An additional promising therapeutic alternative is immune plasma from convalescent patients [7]. This strategy has been used with some success in other viral diseases with significant lethality such as hantavirus, influenza, SARS-CoV, and MERS-CoV infections [8–11].

The use of convalescent plasma for COVID-19 was reported early in this pandemic. The initial case series studies suggested faster clinical recovery, viral clearance, and radiological improvement, although the lack of a control group limited the accurate interpretation of these results [12–14]. Subsequently, a preliminary report of a matched controlled study showed that convalescent plasma improved survival for non-intubated patients [15]. However, the first 2 randomized controlled trials showed no clear clinical benefit, and, furthermore, 1 of these trials was stopped early due to concerns based on finding high preexisting SARS-CoV-2 neutralizing antibody (NAb) titers in patients before transfusion [16,17].

Considering that COVID-19 likely involves at least 2 phases—an early phase in which viral replication is a component of tissue injury and a later phase in which a dysregulated and pro-

inflammatory immune response leads to the damage—the most useful therapeutic window for convalescent plasma administration is currently unknown [18]. Indeed, the lack of efficacy in previous studies has been attributed to a late timing of plasma administration in the disease's course. This hypothesis is consistent with the recent finding of lower mortality for patients receiving convalescent plasma within the first 3 days after COVID-19 diagnosis in a large uncontrolled study [19].

The objective of this study was to assess the efficacy and safety of convalescent plasma therapy in reducing disease progression, complications, and death in patients in the early phase of COVID-19.

## Methods

This study consisted of a randomized, controlled, open-label phase II trial done in a single Chilean academic medical center in Santiago, Chile. Patients were randomized from May 10, 2020, to July 18, 2020, with follow-up until August 17, 2020.

### Inclusion criteria

Inclusion criteria for patients were the following: (1) over 18 years old; (2) hospitalized, with COVID-19 symptoms present at enrollment and confirmed with a positive SARS-CoV-2 real-time PCR in nasopharyngeal swab, or pending PCR result and with imaging consistent with COVID-19 pneumonia and confirmed COVID-19 close contact; (3) ≤7 days from COVID-19 symptom onset to enrollment; (4) a CALL score ≥ 9 points at enrollment (predicts high risk of progression into respiratory failure, based on age, comorbidities, lactate dehydrogenase [LDH], and lymphocyte count) [20]; and (5) Eastern Cooperative Oncology Group (ECOG) performance status before SARS-CoV-2 infection 0–2.

Inclusion criterion number 2 considered initially only SARS-CoV-2 confirmed PCR positive infections. Based on the 24- to 48-hour delays for PCR results in the peak of the pandemic, this criterion was modified after the trial initiation to allow the inclusion of patients with pending PCR test results ($n = 2$) had subsequently confirmed real-time PCR SARS-CoV-2 infection.

### Exclusion criteria

Exclusion criteria were the following: (1) $PaO_2/FiO_2 < 200$ or need for mechanical ventilation at enrollment; (2) coinfection with other relevant respiratory pathogens on admission; (3) pregnancy or lactation; (4) known IgA nephropathy or IgA deficiency; (5) previous immunoglobulin or plasma administration within the last 60 days; (6) previous severe transfusion reactions; (7) do not resuscitate indication; (8) participating in another COVID-19 interventional study; and (9) having, under investigator criteria, any condition that made them unsuitable for study participation.

### Convalescent plasma donation protocol

Plasma was obtained from volunteer participants who had recovered from SARS-CoV-2 infection, having been asymptomatic for at least 28 days, with a negative SARS-CoV-2 real-time PCR both in nasopharyngeal swab and in plasma, and anti-SARS-CoV-2 (S1) IgG titer ≥ 1:400. Donors were males, females who had never been pregnant, or females who had been tested for anti-HLA antibodies. Most of the donors (91%) had a history of symptomatic COVID-19, of which 5% had been hospitalized, but none with severe disease. Plasma collection occurred between 33 and 73 days after symptoms resolved (mean of 44 days). Donor

plasma was tested for standard infectious diseases before administration, and extracted plasma was immediately frozen at −20˚C according to standard national safety measures [21].

Given that for the Euroimmun SARS-CoV-2 IgG ELISA, a positive assay (defined as a ratio of sample optical density [OD]/calibrator OD $\geq$ 1.1) is determined—as per the provider—with a basal dilution of 1:100, we decided to further semi-quantify the IgG in donor plasma with an additional fourth fold dilution, and established the 1:400 cutoff as the requirement for our plasma donors (considering again an OD ratio $\geq$ 1.1 as a positive result for that new dilution).

## Randomization and intervention

Eligible patients were randomly assigned via computer-generated numbering by a block randomization sequence into 2 groups: early or deferred plasma transfusion. Randomization was done by an independent researcher, and the sequence was concealed from study investigators.

The early plasma group received the first plasma unit at enrollment. The deferred plasma group received convalescent plasma only if a prespecified worsening respiratory function criterion was met during hospitalization ($PaO_2/FiO_2 < 200$) or if the patient still required hospitalization for symptomatic COVID-19 >7 days after enrollment.

Transfusions consisted of a total of 400 ml of ABO compatible convalescent plasma, infused as two 200-ml units, each separated by 24 hours. In both groups, cointerventions, including antibiotics, antivirals, heparin thromboprophylaxis, and immunomodulators, were allowed based on the hospital protocols.

## Outcomes

The primary outcome was a composite of mechanical ventilation, hospitalization > 14 days, or in-hospital death.

Secondary outcomes included the following: days of mechanical ventilation, days of high-flow nasal cannula (HFNC), days of oxygen requirement, time to respiratory failure development ($PaO_2/FiO_2 < 200$), the severity of multiple organ dysfunction (by Sequential Organ Failure Assessment [SOFA] score) at day 3 and 7, days in ICU or intermediate care unit, hospital length of stay, and mortality at 30 days. The kinetics of inflammatory biomarkers, including total lymphocyte count, C-reactive protein (CRP), procalcitonin, LDH, D-dimer, ferritin, IL-6, pro-B type natriuretic peptide (pro-BNP), and troponin T, were determined on days 0, 3, and 7, and SARS-CoV-2 real-time PCR in nasopharyngeal swab on days 3 and 7.

Radiological outcomes included the comparison of infiltrate progression on chest CT scans at enrollment and day 5, based on COVID-19 pneumonia severity scores [22–25]. For the combined analysis with portable chest X-rays, a blinded thoracic radiologist expert categorized images as "progression" versus "stable or improved."

Also, preplanned analyses of NAb titers and anti-SARS-CoV-2 IgG titers were conducted in participants from the early plasma group at baseline and in the subset of participants from the deferred plasma group who had not yet received plasma on days 0, 3, and 7.

Analysis of the primary outcome and clinical secondary outcomes was performed by intention to treat (ITT). Laboratory and radiology secondary outcomes were analyzed by modified ITT, excluding a patient who withdrew consent before any intervention. Safety outcomes were evaluated in all participants.

## Anti-SARS-CoV-2 IgG ELISA

For specific IgG enzyme-linked immunosorbent assays (ELISA), we used the commercial kit CE-marked Euroimmun (Lübeck, Germany, # EI 2606–9601 G), which uses the S1-domain of

spike protein of SARS-CoV-2 as antigen. Fresh or thawed serum samples were first diluted at 1:100, immunoreactivity was measured at an OD of 450 nm, and results were expressed according to the manufacturer, with a positive result as an OD ratio (patient/calibrator) $\geq 1.1$. Additionally, 2-fold serial dilutions were done until 1:6,400, and the endpoint dilution for each sample was determined as the final dilution where the OD ratio (patient/calibrator) was $\geq 1.1$. Seroconversion was defined as seronegative at baseline and seropositive after 3 or 7 days, or a 4-fold increase in endpoint dilution titer from the baseline.

### NAb titer assay

Anti-SARS-CoV-2 NAbs were measured in serum samples using an HIV-1 backbone expressing firefly luciferase as a reporter gene and pseudotyped with the SARS-CoV-2 spike glycoprotein [26,27]. Samples with a neutralizing activity of at least 50% at a 1:160 dilution were considered positive and used to perform titration curves and 50% inhibitory dose (ID50) neutralization titer calculations [28]. Determination of the ID50 was performed using a 4-parameter nonlinear regression curve fit measured as the percent of neutralization determined by the difference in average relative light units (RLUs) between test wells and pseudotyped virus controls. In order to perform the ID50 calculations, the lack of fit test had to have a $p$-value $> 0.1$. The top values were constrained to 100, and the bottom values were set to 0.

### Statistical analysis

Sample size was calculated a priori, with a power of 80% and a statistical significance of 5% for an expected outcome of 54.8% of the patients in the control group and 20% in the intervention group experiencing the composite primary outcome (absolute risk reduction of 35%), based on a previous report of convalescent plasma administration in the early stage of AH1N1 influenza [29]. The final calculated sample size was 29 individuals per group (total $n = 58$).

The primary and secondary binary outcomes were assessed through Fisher's exact test, and odds ratios (ORs) are presented together with 95% CIs and $p$-values. Results of all main analyses are presented as crude analyses. In addition, we adjusted for age and SOFA score at enrollment, as fixed (individual-level) effects, using logistic regression. Numerical variables of secondary outcomes were examined using generalized linear models with log link function and gamma family function. For those variables with a high number of zeros, we used a zero-inflated negative binomial model because it showed better goodness of fit compared with other zero-inflated models according to the Akaike information criterion. Treatment effect estimates, crude and adjusted by age and SOFA, are presented as exponentiated coefficients, i.e., ORs and incidence rate ratios (IRRs), respectively, with their corresponding 95% CIs. In those cases where asymptotic assumptions did not hold, crude estimates were analyzed with Fisher´s exact test for categorical variables and Wilcoxon rank-sum test for continuous variables. To test differences between Kaplan–Meier estimates in survival analysis, we used the log-rank test.

For paired CT scan score analysis, we used Wilcoxon matched-pairs signed-rank test.

For the primary endpoint, statistical significance was defined using a 2-sided significance level of $\alpha = 0.05$. The statistical analysis of secondary endpoints should be considered exploratory only. The statistical analysis was performed by an investigator who was blind to the study group allocation. Analyses were done with R version 3.6.3, and figures with GraphPad Prism version 8.4.3 software.

CONSORT guidelines for reporting randomized controlled trials were followed [30]. The trial protocol (S1 Text) and CONSORT checklist (S1 Table) are included for reference.

### Ethics

This study was approved by the institutional review board of the Pontificia Universidad Católica de Chile. Written informed consent was solicited from all patients or their legal representatives.

## Results

### Study population

Of the 245 patients diagnosed with COVID-19 and evaluated for eligibility, a total of 58 patients were enrolled, and 57 (98.3%) completed the trial (1 patient withdrew consent). All patients were included in the ITT analysis (Fig 1). The mean age was 65.8 years (range: 27–92), and 50% were women. The median interval between symptom onset and randomization was 6 days (IQR 4–7). All patients had SARS-CoV-2 infection confirmed by real-time PCR in nasopharyngeal swab. Baseline characteristics of participants are described in Table 1.

All participants ($n = 28$) from the early plasma group received a first plasma unit on the day of enrollment, and 24 (86%) received a second unit 24 hours later. Reasons for not receiving the second unit were death ($n = 2$) or a serious adverse event (SAE) after the first plasma unit administration ($n = 2$).

A total of 13 participants (43.3%) from the deferred plasma group received plasma, at a median time of 3 days from enrollment (IQR 1–5), based on respiratory failure development ($n = 12$) or persistent symptomatic COVID-19 beyond 7 days after enrollment ($n = 1$).

### Primary outcome

There was no significant difference between the early and deferred plasma group in the composite primary outcome: 32.1% (9/28) in the early plasma group versus 33.3% (10/30) in the deferred plasma group (OR 0.95, 95% CI 0.32–2.84). When the outcome was disaggregated, the differences were 17.9% (5/28) versus 6.7% (2/30) (OR 3.04, 95% CI 0.54–17.17) for in-hospital death, 17.9% (5/28) versus 6.7% (2/30) (OR 3.04, 95% CI 0.54–17.17) for mechanical ventilation, and 21.4% (6/28) versus 30.0% (9/30) (OR 0.64, 95% CI 0.19–2.10) for hospitalization > 14 days in the early versus deferred plasma group, respectively (Table 2).

### Secondary outcomes

A total of 46.4% of early plasma group participants progressed to severe respiratory failure ($PaO_2/FiO_2 < 200$) compared to 40% of patients from the deferred plasma group (OR 1.30, 95% CI 0.48–3.56), at a median time of 2.0 and 2.5 days from enrollment, respectively. No significant differences were noted in any of the other clinical secondary outcomes (Table 2). In the adjusted models, the total number of days on mechanical ventilation was higher in the early plasma than in the deferred plasma group (IRR 4.78, 95% CI 2.20–10.40). Time to death and time to severe respiratory failure did not differ between study groups (Fig 2).

No significant differences were found for CRP, IL-6, ferritin, LDH, D-dimer, pro-BNP, troponin T, procalcitonin, and lymphocyte count levels on day 3 and 7 between study groups (S2 Table). Similarly, the rate of SARS-CoV-2 negative real-time PCR in nasopharyngeal swabs did not differ between the early and deferred plasma groups on day 3 (26% versus 8%, $p = 0.204$) nor on day 7 (38% versus 19%, $p = 0.374$) (Fig 3A). As a post hoc analysis, we determined the changes in SARS-CoV-2 PCR cycle thresholds for early plasma group and the subset of patients from the deferred plasma group that did not receive plasma, and we also did not find significant differences (Fig 3B).

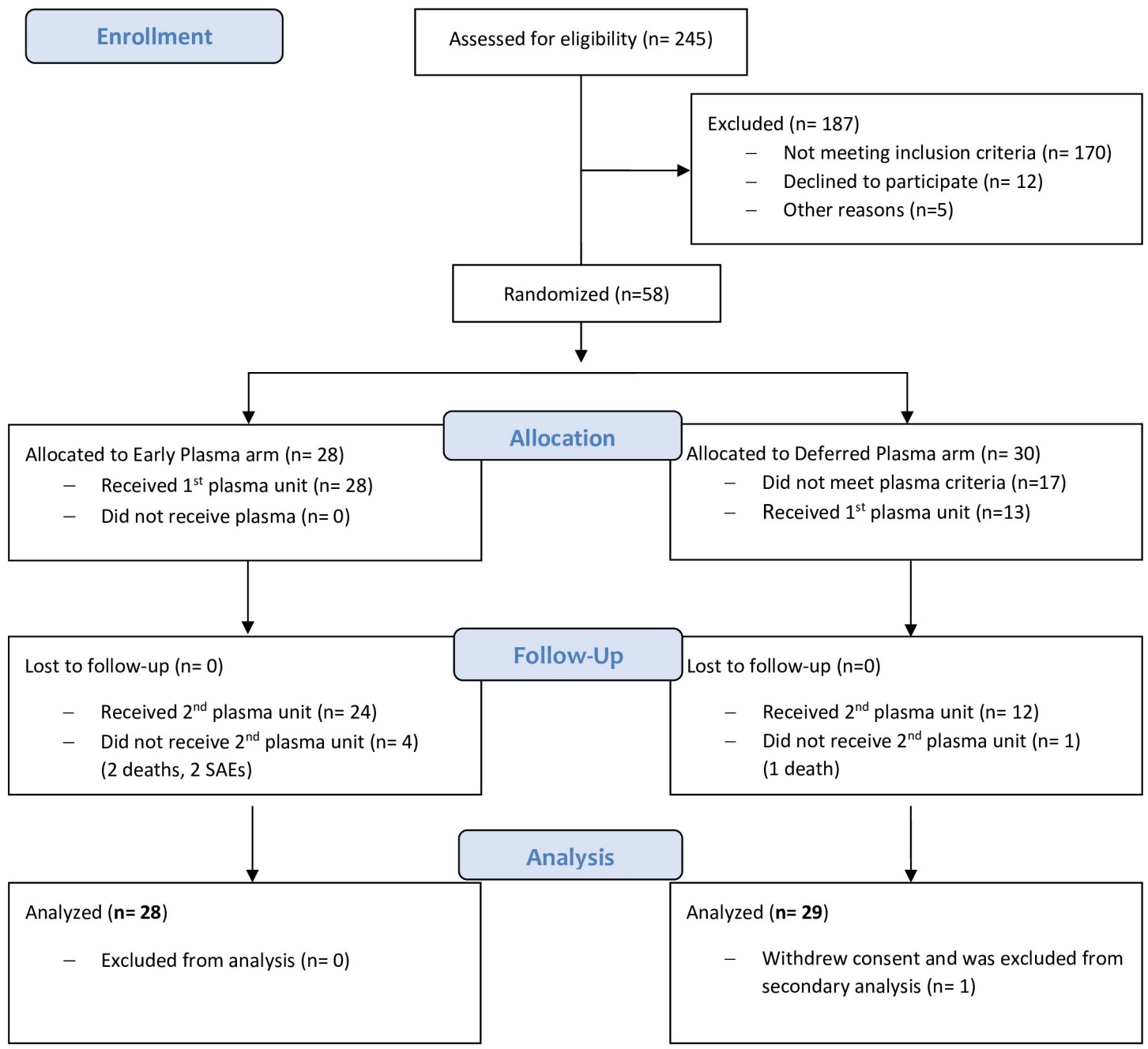

**Fig 1. Study flow diagram.** Patient enrollment and treatment assignment. SAE, serious adverse event.

The progression in the COVID-19 pneumonia (chest CT) severity scores from baseline to day 5 was higher in the deferred than in the early plasma group (S1 Fig). However, when the analysis also included the patients who had a chest X-ray instead of CT on the same scheduled days, the proportion of participants with progression in lung infiltrates did not differ between groups (OR 1.3, 95% CI 0.41–3.89) (S3 Table).

Patients who died ($n = 7$) were older (81 versus 63 years old, $p < 0.001$) and had a higher SOFA score on admission (score of 4 versus 2, $p = 0.002$) than patients who survived ($n = 51$).

**Table 1. Baseline demographics and clinical characteristics of study participants.**

| Characteristic | Early plasma group ($n = 28$) | Deferred plasma group ($n = 30$) |
|---|---|---|
| Age (years), mean (range) | 64.3 (33–92) | 67.1 (27–91) |
| Male sex, number (%) | 15 (53.6) | 14 (46.7) |
| Blood group A, number/total (%) | 7/28 (25.0) | 7/27 (25.9) |
| Blood group O, number/total (%) | 19/28 (67.9) | 14/27 (51.8) |
| Obesity (BMI > 30 kg/m$^2$), number (%) | 3 (10.7) | 4 (13.3) |
| Diabetes mellitus, number (%) | 10 (35.7) | 11(36.7) |
| Hypertension, number (%) | 17 (60.7) | 22 (73.3) |
| Cerebrovascular disease, number (%) | 3 (10.7) | 0 (0) |
| Cancer, number (%) | 1 (3.6) | 3 (10.0) |
| Immunosuppressants, number (%) | 4 (14.3) | 3 (10.0) |
| Chronic renal failure number (%) | 2 (7.1) | 3 (10.0) |
| Chronic liver disease, number (%) | 3 (10.7) | 0 (0.0) |
| Asthma, number (%) | 1 (3.6) | 2 (6.7) |
| Days since COVID-19 symptom initiation, median (IQR) | 5 (4–7) | 6 (4–7) |
| CALL score at enrollment[a], median (IQR) | 10.5 (10–12) | 10.0 (10–12) |
| SOFA score at enrollment[b], median (IQR) | 2.0 (2.0–4.0) | 2.0 (2.0–2.0) |
| O$_2$ requirement at enrollment, number (%) | 23 (82.1) | 23 (76.7) |
| FiO$_2$ requirement at enrollment, median (IQR) | 0.28 (0.22–0.30) | 0.24 (0.21–0.28) |
| PaO$_2$/FiO$_2$ at enrollment, median (IQR) | 260.7 (211–316) | 260.7 (222–308) |
| Lung infiltrates in CT scan or chest X-ray, number (%) | 28 (100.0) | 29 (96.7) |
| **Baseline chest CT severity score** | | |
| Criterion 1[c], median score (IQR) (number) | 18.0 (11.5–26.0) (25) | 14.0 (10.0–19.0) (26) |
| Criterion 2[d], median score (IQR) (number) | 20.0 (15.5–28.0) (25) | 18.0 (15.8–22.5) (26) |
| Criterion 3[e], median score (IQR) (number) | 13.0 (10.0–17.5) (25) | 11.0 (9.0–13.0) (26) |
| Severe pneumonia on CT (severity score > 19, criterion 2[d]), number/total (%) | 14/25 (56.0) | 9/26 (34.6) |
| Severe pneumonia on CT (severity score ≥ 12, criterion 3[e]), number/total (%) | 17/25 (68.0) | 12/26 (46.1) |
| **Other pharmacological interventions for COVID-19 during hospitalization** | | |
| Steroids, number (%) | 23 (82.1) | 20 (66.7) |
| IL-6 blocker (tocilizumab), number (%) | 1 (3.6) | 1 (3.3) |
| Hydroxychloroquine, number (%) | 2 (7.1) | 5 (16.7) |
| Lopinavir/ritonavir, number (%) | 1 (3.6) | 0 (0.0) |
| Thromboprophylaxis[f], number (%) | 25 (89.3) | 23 (76.7) |
| Anticoagulation[f], number (%) | 2 (7.1) | 6 (20.0) |

[a]CALL score (risk of progression into respiratory failure, based on age, comorbidities, lactate dehydrogenase, and lymphocyte count). Reference: Ji et al. [20].

[b]Sequential Organ Failure Assessment (SOFA) score.

[c]Chest CT severity score 1. Reference: Zhou et al. [22].

[d]Chest CT severity score 2. Reference: Yang et al. [25].

[e]Chest CT severity score 3. References: Pan et al. [23,24] and Raoufi et al. [31].

[f]Unfractionated or low-molecular-weight heparin.

Also, median procalcitonin (0.87 versus 0.12 mcg/l, $p = 0.021$), IL-6 (236 versus 41 pg/ml, $p = 0.001$), pro-BNP (3,462 versus 125 ng/l, $p < 0.001$), and troponin T (32 versus 8.6 pg/ml, $p < 0.001$) at enrollment were significantly higher in patients who died at follow-up than in patients who survived.

**Table 2. Primary and secondary clinical outcomes.**

| Outcome | Early plasma group (*n* = 28) | Deferred plasma group (*n* = 30) | *p*-Value[a] | Crude effect estimate (95% CI) | Adjusted effect estimate (95% CI) |
|---|---|---|---|---|---|
| **Primary clinical outcomes** | | | | | |
| Composite outcome (death, mechanical ventilation, and/or hospital stay > 14 days), number/total (%) | 9/28 (32.1) | 10/30 (33.3) | >0.999 | OR 0.95 (0.32–2.84) | OR 0.67 (0.14–3.31) |
| Mechanical ventilation, number/total (%) | 5/28 (17.9) | 2/30 (6.7) | 0.246 | OR 3.04 (0.54–17.2) | OR 2.98 (0.41–21.57) |
| Death, number/total (%) | 5/28 (17.9) | 2/30 (6.7) | 0.246 | OR 3.04 (0.54–17.2) | OR 4.22 (0.33–53.57) |
| Hospitalization > 14 days, number/total (%) | 6/28 (21.4) | 9/30 (30.0) | 0.554 | OR 0.64 (0.19–2.1) | OR 0.51 (0.13–2.05) |
| **Secondary clinical outcomes** | | | | | |
| 30-day mortality, number/total (%) | 5/28 (17.9) | 2/30 (6.7) | 0.246 | OR 3.04 (0.54–17.2) | OR 4.22 (0.33–53.57) |
| Progression into respiratory failure[b], number/total (%) | 13/28 (46.4) | 12/30 (40.0) | 0.791 | OR 1.30 (0.46–3.68) | OR 1.46 (0.43–4.66) |
| Total days of mechanical ventilation requirement, median (IQR) | 0.0 (0.0–0.0) | 0.0 (0.0–0.0) | 0.234 | IRR 1.68 (0.30–9.42) | IRR 4.78 (2.20–10.40) |
| Total days of HFNC requirement, median (IQR) | 0.0 (0.0–2.5) | 0.0 (0.0–2.0) | 0.751 | IRR 0.70 (0.35–1.43) | IRR 0.65 (0.35–1.30) |
| Total days of oxygen requirement, median (IQR) | 6.0 (3.0–12.0) | 7.0 (2.0–16.0) | 0.950 | IRR 0.90 (0.53–1.53) | IRR 1.07 (0.64–1.78) |
| Total days of intensive and/or intermediate care requirement, median (IQR) | 2.5 (0.0–8.25) | 0.0 (0.0–8.5) | 0.438 | IRR 0.69 (0.37–1.31) | IRR 0.68 (0.36–1.26) |
| Total days of hospital stay, median (IQR) | 9.0 (5.0–12.0) | 8.0 (5.5–23.0) | 0.806 | IRR 0.78 (0.50–1.22) | IRR 0.86 (0.57–1.29) |
| SOFA score day 3, median (IQR) | 2.0 (1.0–4.0) | 2.0 (1.0–3.0) | 0.728 | IRR 1.18 (0.78–1.79) | IRR 1.12 (0.84–1.48) |
| SOFA score day 7, median (IQR) | 2.0 (2.0–4.0) | 3.0 (1.0–4.0) | 0.565 | IRR 1.29 (0.74–2.22) | IRR 0.98 (0.65–1.48) |

Adjusted ORs were estimated from a logistic regression model, and IRRs were estimated using a zero-inflated negative binomial model. Estimates were adjusted by age and SOFA score at enrollment.

[a]*p*-Value was calculated by Wilcoxon rank-sum test or Fisher's exact test.

[b]Respiratory failure defined as $PaO_2/FiO_2 < 200$.

HFNC, high-flow nasal cannula; IRR, incidence rate ratio; OR, odds ratio; SOFA, Sequential Organ Failure Assessment.

## Immune response subgroup analysis

From a total of 232 potential plasma donors with baseline positive SARS-CoV-2 IgG detection, 129 candidates (55.6%) achieved the further positive cutoff at the 1:400 dilution. The median

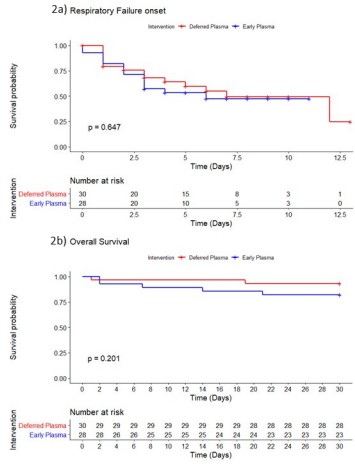

**Fig 2. Time from enrollment to key secondary outcomes.** (A) Time from enrollment to severe respiratory failure development ($PaO_2/FiO_2 < 200$) in the early plasma and deferred plasma groups. (B) Time from enrollment to death in the early plasma and deferred plasma groups.

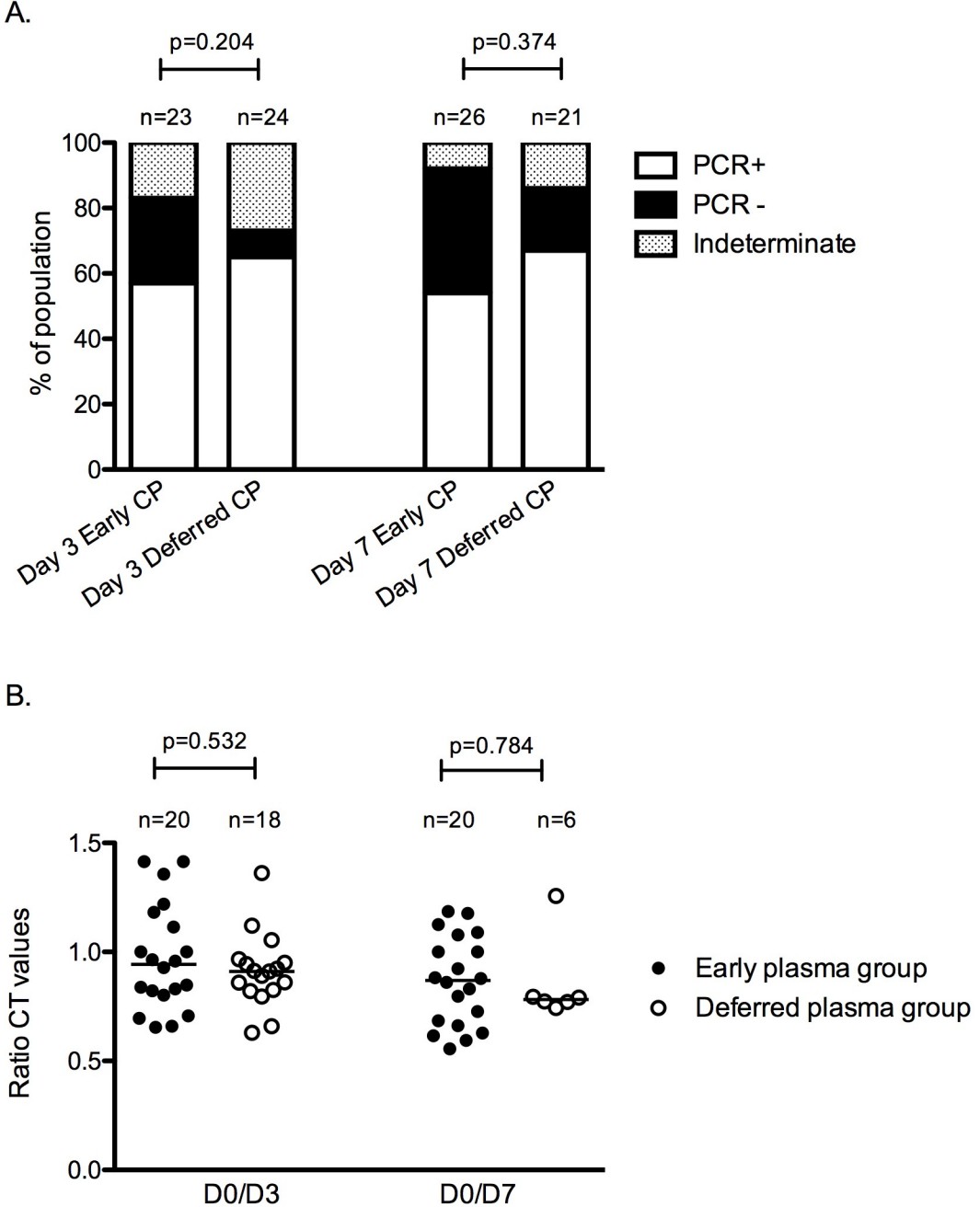

**Fig 3. SARS-CoV-2 real-time PCR in nasopharyngeal samples.** (A) Each column represents the real-time PCR results obtained in patients from the early plasma group and deferred plasma group. Above the columns, the number of samples is indicated. The proportion of positive PCR is represented in white, negative PCR in black, and an indeterminate result (CT ≥ 35) dotted. (B) Changes in real-time PCR CT for patients from the early plasma group and for the subset of patients from the deferred plasma group that did not receive plasma before day 3 or day 7. Results are expressed as the ratio between CT values from day 0 (D0) to day 3 (D3) and D0 to day 7 (D7). Filled circles represent each patient from the early plasma group, and open circles represent patients from the deferred plasma group. Above the scatter plots, the number of available samples is indicated; lines represent the medians. CP, convalescent plasma; CT, cycle threshold.

SARS-CoV-2 IgG OD ratio at the standard basal dilution (1:100) for all donors ($n = 41$) whose plasma was administered to the patients in this clinical trial was 5.73 (IQR 4.73–6.51). Additionally, in 18 of the 41 (44%) plasma donors, the virus neutralizing capacity was measured, and the median titer of NAb ID50 was 449 (range: 147–5,610). The baseline SARS-CoV-2 IgG ratio in all donors ($n = 28$) whose plasma was given to the early plasma group patients, versus IgG ratio in all donors ($n = 13$) whose plasma was administered to the deferred plasma group patients, was not different (median IgG OD ratio—at standard basal 1:100 dilution—of 5.77 and 5.73, respectively, $p = 0.808$).

SARS-CoV-2 IgG levels were determined in patients who received early plasma and in the subset of patients from the deferred plasma group who had not yet received plasma, at baseline, day 3, and day 7. No significant differences were observed in SARS-CoV-2 IgG seropositive rate at any of the 3 timepoints (Fig 4A). Regarding IgG titers at enrollment, 7/26 (27%) of patients who subsequently received plasma had a positive SARS-CoV-2 IgG assay, with a median IgG titer of 400 (range: 100–800), compared to 5/20 (25%) of those patients who did not receive plasma, with a median IgG titer of 400 (range: 100–3,200), a non-significant difference in titers ($p = 0.548$). On day 3, 19/26 (73%) of patients who received plasma had a positive SARS-CoV-2 IgG assay, with a median IgG titer of 400 (range: 100–3,200), compared to 10/20 (50%) of those who had not yet received plasma, with a median IgG titer of 400 (range: 100–3,200), a non-significant difference in titers ($p = 0.962$). Also, no significant differences were observed in IgG seroconversion rates between those who received plasma and those who received no plasma at day 3 (69% versus 40%, $p = 0.073$) or at day 7 (87% versus 83%, $p = 1.00$) (Fig 4B).

None of the 7 patients who died had positive SARS-CoV-2 IgG at enrollment (median OD ratio 0.3, IQR 0.1–0.4).

Additionally, NAbs were quantified for a total of 44 study patients: At enrollment, 59% (26/44) did not reach the screening cutoff (50% neutralization) at the 1:160 dilution (Fig 4C). Interestingly, only 16% (3/19) of patients enrolled within 5 days after COVID-19 symptom onset had ID50 titers $\geq$ 1:160, compared to 60% (15/25) of those enrolled on day 6 or 7 after symptom onset ($p = 0.005$) (Fig 4D).

## Safety

Among the 41 patients receiving plasma in this study, there were 4 possibly related adverse events (3 cases of fever, 1 rash) and 3 SAEs (7.3%). Two patients developed severe respiratory deterioration within 6 hours after plasma infusion and were categorized as having possible transfusion-associated acute lung injury (TRALI) type II [32]. One of these patients additionally developed severe thrombocytopenia within 48 hours after plasma transfusion, with megakaryocytic hyperplasia in the bone marrow analysis. Platelet antibody testing in the recipient was negative, as well as in the donor plasma, ruling out passive alloimmune thrombocytopenia. Platelet count remained low in the following weeks, despite platelet transfusions, steroids, and immunoglobulin therapy, with the patient requiring splenectomy, rituximab, and eltrombopag before slow stabilization. This event was diagnosed as a complication possibly related to COVID-19.

## Discussion

This randomized clinical trial of symptomatic COVID-19 patients admitted early failed to find significant differences in the composite primary outcome of death, mechanical ventilation, or prolonged hospitalization between administering immediate convalescent plasma and administering plasma only in case of clinical worsening.

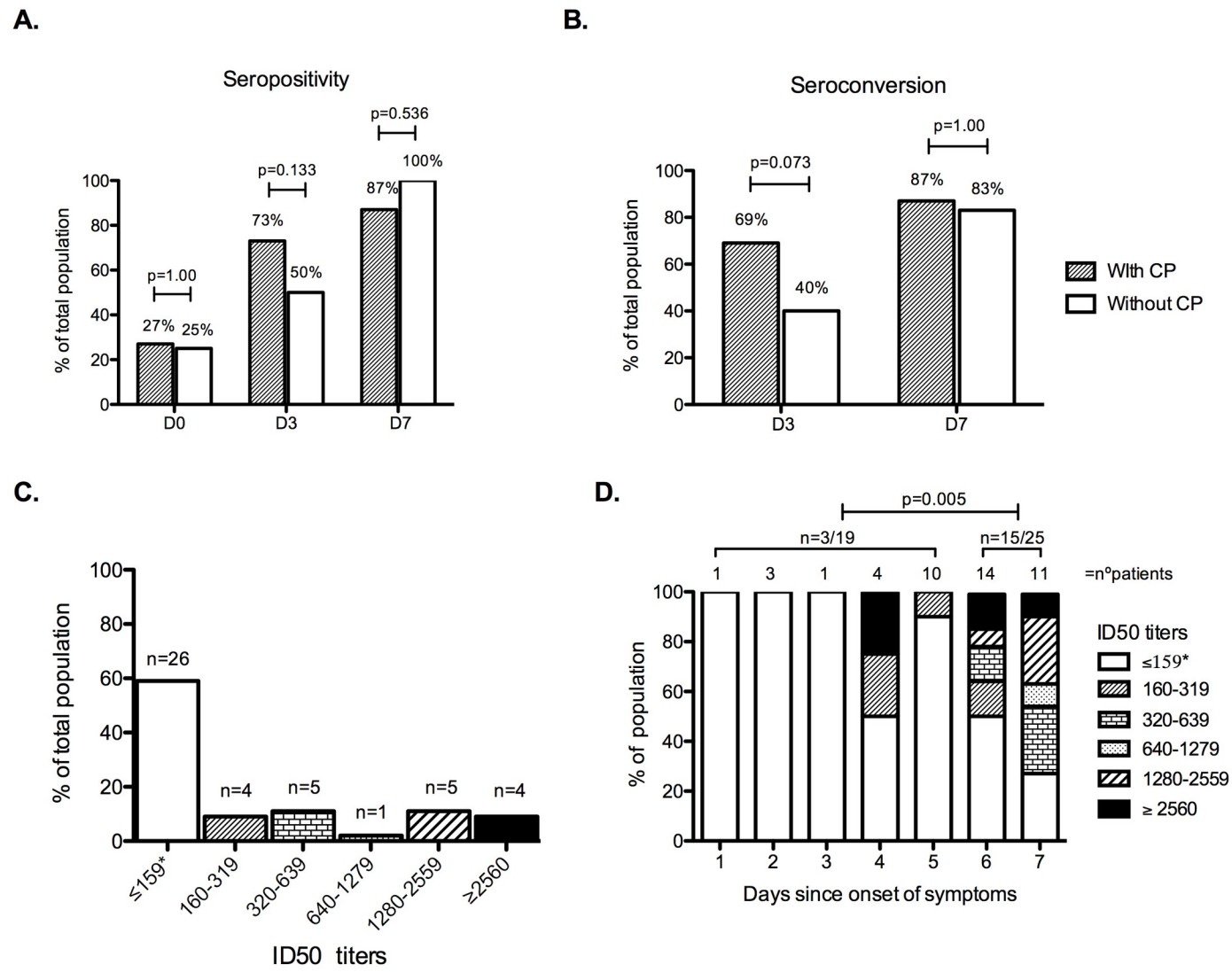

**Fig 4. The humoral immune response induced by SARS-CoV-2.** (A) IgG seropositivity (optical density [OD] ratio ≥ 1.1) analysis by sampling day: D0, day of enrollment; D3, the third day after enrollment; D7, 7 days after enrollment. Dashed columns represent the patients who received convalescent plasma (CP) at enrollment ($n$ = 26 samples available on day 0, $n$ = 26 on day 3, and $n$ = 23 on day 7); white columns represent the patients from the deferred plasma group who did not receive plasma ($n$ = 20 samples available on day 0, $n$ = 20 on day 3, and $n$ = 12 on day 7). Above each column, the percentage of seropositivity is indicated. (B) IgG seroconversion was considered to have occurred if a patient had a negative sample at 1:100 dilution at baseline but increased to any positive dilution after 72 hours or 7 days, or if a 4-fold increase in endpoint dilution titer from baseline was reached. Dashed columns represent patients who received CP; white columns represent patients from the deferred plasma group who did not receive plasma. Above each column, the percentage of seroconversion is indicated. (C) Neutralizing antibody (NAb) titer measured by 50% inhibitory dose (ID50) quantified at D0. The total number of patients reaching every dilution titer interval is indicated above each column. *ID50 titer ≤ 1:159 or no neutralization observed. (D) NAb titers showed by days since COVID-19 symptom onset. Each column represents the number of days after onset of symptoms; above each column are the number of individuals. Summary statistics above represent the number of individuals with NAb titers ≥ 1:160 from 2 groups: those enrolled in the first 5 days since symptom onset and those enrolled 6 or 7 days since symptom onset. *ID50 titer ≤ 1:159 or no neutralization observed.

The rate of SARS-CoV-2 PCR clearance in nasopharyngeal swabs did not differ between study arms either, suggesting that the provision of convalescent plasma in this study did not provide enough antiviral activity in patients with COVID-19 at this stage. In accordance with this finding, transfused patients did not present a significant rise of SARS-CoV-2 IgG levels on

days 3 and 7 compared to the natural increase in IgG titers in non-infused patients, which could explain a possible lack of effect. Furthermore, almost 30% of patients were still not sero-positive at 72 hours after the infusion, which suggests that the volume of infused plasma or its antibody concentration may have been insufficient.

We actively selected patients at high risk of developing complications—based on CALL score—and indeed, over 40% of our participants developed severe respiratory failure. The failure to find clinical benefit from convalescent plasma therapy in these patients may be explained by several reasons. First, humoral immunity may not play a major role in the subset of patients who have already initiated a highly pro-inflammatory response and in whom inflammation and coagulopathy may be more important than viral replication in disease progression [33]. We do not know whether preselection of plasma units with a very high concentration of NAbs or a larger volume of plasma could have succeeded in blunting this dysregulated inflammatory response. Additionally, an early adaptive immune response might be necessary to drive more effective infection control. Indeed, different cellular and humoral responses are generated in mild versus severe COVID-19 cases, and it has been reported that a specific cellular response can be detected early in the course of non-severe COVID-19 [34,35]. Second, the possible lack of efficacy may relate to a too late administration of plasma in the course of the disease, in which a dysregulated immune response predominates and is independent of the virus cell entry blockade achieved by immunoglobulins [7,18]. Previous randomized trials of convalescent plasma for COVID-19 included patients who had a longer time gap between symptom onset and transfusion as well as more severe disease at enrollment [16,17]. Despite setting a strict ≤7 days of symptoms inclusion criterion, in our study, over 96% of participants had already established pneumonia on CT scan at enrollment. Hence, it is possible that some patients had a more rapid or aggressive course, or, particularly for older adults, true COVID-19 symptom onset went unnoticed until several days into the disease course. Nonetheless, that the study population reflected early-stage COVID-19 was well supported by the fact that at enrollment over 74% of our participants did not have detectable SARS-CoV-2 IgG, and about 60% did not have significant NAb capacity. Third, given the design of our study, plasma administration in the deferred plasma group may have prevented the primary outcome from developing. However, the probability and time to progression to respiratory failure did not differ between the study groups. Since respiratory failure was one of the prespecified criteria for plasma administration in the deferred plasma group, this secondary outcome allowed us to compare early plasma versus no plasma, further supporting a possible lack of efficacy.

Plasma transfusion is not exempted from adverse events such as allergic reactions, infection transmission, and—very rarely—volume overload or TRALI [36]. In spite of the fact that the majority of clinical trials of convalescent plasma in COVID-19 were still ongoing, convalescent plasma received emergency use authorization by the US Food and Drug Administration for the treatment of hospitalized patients with COVID-19 [37]. Reassuringly, in a recent report of 20,000 hospitalized patients receiving convalescent plasma for COVID-19, the incidence of related SAEs in the first 4 hours after infusion was <0.5% [38]. Nonetheless, in the present study, 2 participants developed acute respiratory failure after transfusion. Given that the patients were, according to the known evolution of COVID-19, in the peak of their inflammatory phase, it was challenging to determine if the respiratory failure corresponded to a TRALI [39].

Our study presents some limitations. First, NAbs were not determined in donor plasma before the patient's transfusion, and we could not select the plasma units with the highest neutralizing activity. Additionally, there is a critical knowledge gap regarding the dose of convalescent plasma needed to effectively increase the pool of antibodies required to neutralize the virus in the blood and in other compartments, and in the present study, the non-significant

change of antibody titers suggests that the convalescent plasma dose may have been insufficient. Second, the study was not powered to detect a risk reduction smaller than 35% in the primary endpoint, and therefore we cannot exclude that convalescent plasma may show smaller but clinically relevant effects in a future larger clinical trial. Third, as this was an open-label study, cointerventions such as steroid use may have unintentionally influenced outcomes [5]. Such management was not standardized, although alternative drug therapies were equally distributed in both study arms.

Regarding applicability, we found it difficult to find patients admitted to hospital in the early stages of COVID-19. Other large case series have reported that the median time from symptom onset to hospital admission was 7 days in the US and 6 days in Madrid, Spain [40,41]. Thus, a considerable proportion of patients will inevitably have passed the 7-day symptom window when admitted. This implies that new strategies such as outpatient plasma administration in newly diagnosed SARS-CoV-2 infections in selected patients at higher risk of COVID-19 complications—such as older individuals with comorbidities—should be explored. However, before proceeding with further large clinical trials, convalescent plasma dosage (volume and antibody titer levels) critically requires optimization; this could be studied safely in healthy volunteers or in post-exposure prophylaxis studies.

In conclusion, the present clinical trial of convalescent plasma administered in patients hospitalized in the early stage of COVID-19, compared to giving plasma only at clinical deterioration, failed to demonstrate improvement in clinical outcomes. Newer research strategies are needed to find the optimal use and timing of convalescent plasma in COVID-19.

## Supporting information

**S1 Fig. Chest CT COVID-19 pneumonia severity scores at baseline (day 0) and at day 5, for the early convalescent plasma ($n$ = 12) and deferred plasma ($n$ = 18) group.** (A) CT score 1 (Zhou et al. [22]). (B) CT score 2 (Yang et al. [25]). (C) CT score 3 (Pan et al. [23,24]). (TIF)

**S1 Table. CONSORT checklist.** (DOC)

**S2 Table. Laboratory outcomes.** (DOCX)

**S3 Table. Radiological changes (from day 0 to day 5) for the early and deferred plasma groups, based on expert radiologist criteria.** (DOCX)

**S1 Text. Study protocol.** (DOCX)

## Acknowledgments

We would like to thank all plasma donors, who volunteered in great numbers, and all patients who participated in the study; the institutional review board at Pontificia Universidad Católica de Chile for its efficient review of the application; all study nurses and in particular Haylim Nazar, Monique Moreau, Soledad Navarrete, Silvana Llevaneras, Carolina Henriquez, and Camila Carvajal for their generous commitment to this project; the Dirección de Investigación y Doctorados de la Escuela de Medicina UC for its collaboration in study finances management and coordination; our colleagues Teresita Quiroga and Ana María Guzmán for their

collaboration with serologies and general laboratory analysis; and all staff from the Diagnostic Virology Laboratory, Red de Salud UC CHRISTUS.

## Author Contributions

**Conceptualization:** María Elvira Balcells, Luis Rojas, Nicole Le Corre, María Elena Ceballos, Marcela Ferrés, Mayling Chang, Cecilia Vizcaya, Sebastián Mondaca, Jaime Santander, Christian Caglevic, Mauricio Mahave, Raimundo Gazitúa, José Luis Briones, Jaime Pereira, Bruno Nervi.

**Data curation:** Luis Villarroel, Marcela Ferrada.

**Formal analysis:** María Elvira Balcells, Nicole Le Corre, Marcela Ferrés, Sebastián Mondaca, Álvaro Huete, Ricardo Castro, Luis Villarroel, Sergio Vargas-Salas, Carolina Beltrán-Pavez, Ricardo Soto-Rifo, Fernando Valiente-Echeverría, Carlos Balmaceda, Manuel A. Espinoza.

**Funding acquisition:** María Elvira Balcells, Jaime Santander, Bruno Nervi.

**Investigation:** María Elvira Balcells, Luis Rojas, Nicole Le Corre, Constanza Martínez-Valdebenito, María Elena Ceballos, Mayling Chang, Cecilia Vizcaya, Ricardo Castro, Mauricio Sarmiento, Alejandra Pizarro, Patricio Ross, Bárbara Lara, Carolina Beltrán-Pavez, Ricardo Soto-Rifo, Fernando Valiente-Echeverría, Christian Caglevic, Carolina Selman, Raimundo Gazitúa, José Luis Briones, Jaime Pereira.

**Methodology:** Luis Rojas, Sebastián Mondaca, Alejandra Pizarro, Ricardo Soto-Rifo, Fernando Valiente-Echeverría, Franz Villarroel-Espindola, Carlos Balmaceda, Manuel A. Espinoza.

**Project administration:** Marcela Ferrada.

**Resources:** Jaime Santander, Carolina Selman, Bruno Nervi.

**Supervision:** María Elvira Balcells, Marcela Ferrada, Carolina Selman.

**Validation:** María Elvira Balcells, Sebastián Mondaca, Marcela Ferrada.

**Visualization:** Constanza Martínez-Valdebenito, Sergio Vargas-Salas, Carolina Beltrán-Pavez, Carlos Balmaceda, Manuel A. Espinoza.

**Writing – original draft:** María Elvira Balcells.

**Writing – review & editing:** María Elvira Balcells, Luis Rojas, Nicole Le Corre, Constanza Martínez-Valdebenito, María Elena Ceballos, Marcela Ferrés, Mayling Chang, Cecilia Vizcaya, Sebastián Mondaca, Álvaro Huete, Ricardo Castro, Mauricio Sarmiento, Alejandra Pizarro, Patricio Ross, Bárbara Lara, Ricardo Soto-Rifo, Fernando Valiente-Echeverría, Christian Caglevic, Mauricio Mahave, Raimundo Gazitúa, José Luis Briones, Franz Villarroel-Espindola, Jaime Pereira, Bruno Nervi.

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
