## [Editor Report · Decision Letter 0]

21 Sep 2020

Dear Dr Balcells, 

Thank you for submitting your manuscript entitled "Early Anti-SARS-CoV-2 Convalescent Plasma in Patients Admitted for COVID-19: A Randomized Phase II Clinical Trial" for consideration by PLOS Medicine.

Your manuscript has now been evaluated by the PLOS Medicine editorial staff and I am writing to let you know that we would like to send your submission out for external peer review.

Kind regards,

Artur Arikainen,

Associate Editor

PLOS Medicine

---

## [Decision Letter · Decision Letter 1]

15 Oct 2020

Dear Dr. Balcells,

Thank you very much for submitting your manuscript "Early Anti-SARS-CoV-2 Convalescent Plasma in Patients Admitted for COVID-19: A Randomized Phase II Clinical Trial" (PMEDICINE-D-20-04556R1) for consideration at PLOS Medicine. 

[LINK]

In light of these reviews, I am afraid that we will not be able to accept the manuscript for publication in the journal in its current form, but we would like to consider a revised version that addresses the reviewers' and editors' comments. Obviously we cannot make any decision about publication until we have seen the revised manuscript and your response, and we plan to seek re-review by one or more of the reviewers. 

We expect to receive your revised manuscript by Nov 05 2020 11:59PM. Please email us (plosmedicine@plos.org) if you have any questions or concerns.

We look forward to receiving your revised manuscript. 

Sincerely,

Emma Veitch, PhD

PLOS Medicine

On behalf of Artur Arikainen, PhD

Associate Editor, PLOS Medicine

plosmedicine.org

*In the Abstract "methods and findings" section, please make sure all sentences (eg the below) are full sentences, ie with a verb, rather than sentence fragments: "Open-label, single-center, randomized clinical trial performed in an academic center in Santiago, Chile, from May 10, 2020, to July 18, 2020, with final follow-up August 17, 2020".

*In multiple places in the manuscript (see also below, and highlighted by the reviewers), the authors should rephrase the main conclusions so that it's clear they are not claiming their study has shown lack of benefit but rather failed to find evidence of benefit, given the lack of power on primary outcome: eg Abstract, "We found no benefit in the primary outcome (32.1% vs 33.3%, OR 0.95, 95% CI 0.32-2.84, p>0.99) in the early versus deferred CP group." 

*In the last sentence of the Abstract Methods and Findings section, please include a note summarising key limitation(s) of the study's methodology.

*Per the note above about power issues, the following statement in the abstract should be rephrased accordingly - "Immediate addition of CP therapy in early stages of COVID-19 -compared to its use only in case of patient deterioration- did not confer benefits in mortality" - this would be better read as saying "We failed to find evidence of benefit..."

*At this stage, we ask that you include a short, non-technical Author Summary of your research to make findings accessible to a wide audience that includes both scientists and non-scientists. The Author Summary should immediately follow the Abstract in your revised manuscript. This text is subject to editorial change and should be distinct from the scientific abstract. Please see our author guidelines for more information: https://journals.plos.org/plosmedicine/s/revising-your-manuscript#loc-author-summary

*In the Introduction, in the following sentence, "contagions" would be better phrased as "infections" - "The SARS-CoV-2 pandemic has left over 24 million contagions and 833,000 deaths by August 29, 2020[1]"

*In the Introduction, can you say anything to give context about the types of settings the cited case series pertain to? Eg, high, middle, low-income countries, any specific considerations about the eligibility for these series (inclusion/exclusion criteria - adults, children, specific vulnerabilities, population-representative cohorts etc?) - "Different case-series have shown an intensive care unit (ICU) admission rate between 5 and 16%, and a case fatality rate near 1-4%"

*In the Methods section, can you add a sentence in an appropriate location noting that the trial was reported per the CONSORT guidelines and add a citation there for the CONSORT instrument?

*Per the reviewer's comment, in the Methods section where you detail power considerations, can you add a note saying whether this was an a priori or post-hoc power calculation? Many thanks - "Sample size was calculated with a power of 80% and a statistical significance of 5% for an absolute risk reduction of 35% in the primary outcome"

*In the Discussion, although in some places the conclusions are framed as a failure to find benefit, in other places it's stated as a finding of lack of benefit, eg the following - "The lack of clinical benefit from convalescent plasma therapy in these patients may be explained by several reasons" - in each case this needs to be reframed so the discussion is focussed around the wide CI's for the effects on main outcomes and therefore with the evidence from this trial being consistent with a range of possible effects (including possible benefit). 

*The above point also pertains to the following statement, and perhaps others in the discussion too - "Secondly, limited efficacy may be due to a late administration in the course of the disease..."

*Finally, the final take-home message in the Discussion (below) will also have to be moderated, given the above point - "In conclusion, convalescent plasma transfusion in patients hospitalized in the early stage of COVID-19, compared to giving plasma only at clinical deterioration, did not improve clinical outcomes.." 

Comments from the reviewers:

Reviewer #1: This small RCT on convP for COVID has as the unique aspect that patients were treated within 7d after symptom onset. Although the study is clearly too small to draw definite conclusions on efficacy, it gives some relevant insights into antibody kinetics after transfusion. 

line 77: The authors should briefly mention the evident positive impact of dexam on mortality and the shorter time to recovery with remdesivir

Donors were screened based on an anti-SARS-CoV-2 (S1) IgG titers 1:400 (ELISA Euroimmun®). It is not entirely clear to me how this 1:400 relates to OD ratio (the official measure of the Euroimmun test). To get an indirect idea of the PRNT50 titers in donor plasma the authors should mention the euroimmun OD ratio of the donor serum rather than the titer as the OD ratio correlates well with the SARS2 plaque reduction neutralization essay (see https://www.nature.com/articles/s41467-020-17317-y/figures/1). The authors should also mention what percentage of the donors were rejected based on this 1/400 cut-off. e.g Did this cutoff lead to the use of the 10% highest titers of all donors tested or was it the top 5 or 25%?

Line 163: At 1/101 should probably be 1/10 or 1/100. Please correct

The power calculation cannot be correct in my opinion. Every 35% reduction study has a sample size of hundreds of patients unless one would anticipate that the endpoint occurs in the majority of the controls (which is not the case). Also , a sample size calculation should be completely reproducible by an independent statistician who reads the paper. Therefore, it should mention the exact assumptions and methods used: What was the expected incidence of the prim endpoint in the control arm and the intervention arm. If this was a post-hoc power analysis, it should be removed and the authors could state instead that no power calculation was performed. 

The study had one primary endpoint (composite of death, mech ventil or >14d hospital stay). So only one statistic should have been predefined. Instead, the authors write that the primary binary outcomes were assessed through chi-squared tests or Fisher's exact test when appropriate, and odds ratios (ORs) are presented together with 95% CI and p-values. That is confusing. Which test was used for the primary endpoint? 

Given the small sample size, the number of analyses endpoints is extremely high. I suggest to move most of the secondary endpoints to a online supplement. Even the analysis on the different subgroups of the compos prim endpoint are not very relevant and could be removed or moved to online supplement. The same can be said for the baseline characteristics table. 

Fig3. The positive/negative PCR result analysis is likely to be a very insensitive method because even in a patient who has fully recovered from COVID, PCR can remain positive for weeks to months. A better analysis would be to look at the change in cycle treshold values from d1 to day 3 or from d1 to day 7 and compare this between CP and controls (excluding those from the control group who received rescue Cplasma before day 3). I urge the authors do do this as a post hoc analysis and decribe the results (and mention that it was a post-hoc analysis).

The discussion should very clearly mention that the study can in no way conclude that plasma therapy is ineffective given the small sample size (and the sample size calculation that I assume is incorrect). So the statement "it is possible that the study underpowered" should be removed and be clear about the lack of power to detect a clinically relevant effect size. However, the study does provide relevant information about the knowledge gaps we still have: - Why are not all patients antibody positive on day 3 after the plasma transfusion? Is it because the dosing is inadequate ? Have the antibodies been consumed quickly by attaching to the virus and is a much higher dose or plasma with much higher titers needed (e.g. some studies are using 4 units over 2 days). Has the window of opportunity been past when patients are sick enough to enter the the hospital due to a overwhelming inflammatory respons, even if sick for <8 days? 

A limitation that should be mentioned is that typically +- 75% of the patients will have passed the 7d symptoms window when they are admitted to the hospital. Another limitation that should be mentioned is the lack of any dose finding data on conval plasma in healthy volunteers and the fact that the antibodies in 400 ml of plasma given to an adult patient will be diluted in +-6 times this volume, let alone the consumption of antibodies that leave the blood compartment and/or bind the virus. 

The study should give some short recommendations on how further antibody based therapies (plasma, monoclonal highly neutral antibody therapy) should be studied. Some suggestions are : Focus on high risk patients before admission. First establish what dose is appropriate in H volunteers. Multicenter large trials etc. 

Reviewer #2: This was an open, labelled, single-center, randomized clinical trial conducted at an academic center in Santiago, Chile. It focused on the application of convalescent plasma in COVID-19 patients. The 58 patients included in the study were randomly divided into an early plasma group and a delayed plasma group. The authors found that there was no significant difference in the main efficacy of the early CP group and the delayed CP group. The author believed that compared to using it only when the patient's condition deteriorates, the immediate addition of CP treatment in the early stage of COVID-19 did not bring benefits in terms of mortality, length of hospitalization, or mechanical ventilation needs. The clinical research is very rigorous in design and development. However, there are several concerns in the following:

1. Different cellular and humoral responses are generated in mild or severe COVID-19 cases, and it has been reported that a specific cellular response can be detected early in the course of non-severe COVID-19. So what explains the cellular and humoral responses of the patients included in this study? Why not perform subgroup analysis of mild and severe covid-19 patients?

2. Additional standard treatment was allowed in both arms. What are the main additional standard treatments? For example, does the use of steroids interfere with the conclusions reached?

3. In previous studies, it has been observed that different levels of IgG plasma have a certain impact on patient mortality, and even unadjusted IgG has a dose-response relationship with 30-day mortality. How to ensure the consistency of convalescent plasma IgG collected from volunteers?

4. Different patients have different degrees of progress in the course of the disease. Some elderly people only show real COVID-19 symptoms in the later stages of the disease, and some people have asymptomatic infection. 

5. The progression of COVID-19 pneumonia CT severity score from baseline to day 5 in the delayed treatment group was higher than that in the early plasma group. Perhaps you should continue to study and increase the number of patients enrolled.

Reviewer #3: Balcells et al describes a randomized trial of COVID-19 convalescent plasma (CCP) in COVID-19 infection comparing early and late administration of CCP. A composite endpoint as well as individual secondary outcomes were examined. The authors failed to identify any differences in primary composite outcome and a number of secondary outcomes between high risk of progression patients receiving CCP early and those receiving it only upon progression according to predefined criteria.

Overall, the paper is well written and represents important new information concerning the use of CCP in the ongoing pandemic. I have one major concern which could influence the paper as well as a some minor concerns.

Major Concern

1) On page 5, lines 121 through 126 the COVID convalescent plasma is described including the testing performed to determine whether antibodies to SARS-CoV-2 were present and the titer of the units. It is indicated that the Euroimmun ELISA was used to test the units and that the IgG titers were >1:400. The Euroimmun ELISA, however, is not a qualitative assay capable of determining the titer. Additional information and detail is needed to describe how the titer was determined utilizing this test. For example, in the Expanded Access Protocol recently described from the United States, the Ortho VITROS SARS-CoV-2 IgG assay was compared to the Broad PRNT neutralizing assay to determine the S/C ratio corresponding with the various titers reported. presumably something similar was done. This should be described as the titer of the units could conceivably determine the efficacy of the plasma and therefore the outcomes of the study. Latter it is mentioned that a subset of units were tested to determine the titer of nAb by the authors.

2) On page 17, lines 314 through 316 - The authors describe possible TRALI reactions. The overall reaction rate is much higher than that reported In the Expanded Access Program which has reported on 20,000 patients receiving CCP. Were the donors providing the CCP either male donors, female donors who had never been pregnant, or were the donors screened for the presence of anti-HLA antibodies? This is not stated in the initial description of the CCP. Please indicate whether or not such TRALI risk reduction measures were implemented both in your description of the plasma product as well as in your safety results. 

3) On page 17, lines 316 through 321 the authors describe profound thrombocytopenia following transfusion of CCP in a single patient. They mention that testing for antiplatelet antibodies was negative. Please clarify whether this testing was performed in the recipient or in the donor. If in the recipient, reports of passive alloimmune thrombocytopenia (PAT) have appeared in the literature and in many of these, antibodies were not detected in the recipient despite being present in the transfused blood product. Absence of antiplatelet antibodies in the recipient may not exclude this entity.

Minor

1) Page 16, line 286 - The phrase "plasma receptors" is rather awkward and not correct English. I would suggest rewriting as "...seroconversion rates between those who received plasma and those who received no plasma at day 3..."

Reviewer #4: This is a useful RCT on the efficacy of early Anti-SARS-CoV-2 Convalescent Plasma treatment vs late plasma treatment as needed in Patients Admitted for COVID-19. The statistical methods and analyses and presentation of the results (tables and figures) are mostly adequate. However, there are some major issues needing attention especially on the sample size.

1) Sample size. The sample size of 58 was calculate based on the assumption of an absolute risk reduction of 35% in the primary outcome. This is a huge effect size needed to be carefully justified. The authors used the paper on "Hyperimmune IV Immunoglobulin Treatment" as the reference for this 35% risk reduction, which however could not be found in the paper. The paper itself also acknowledged lack of statistical power due to small sample size, and the disease, treatment arms and primary outcome are very different from those in this paper. Thinking about the results of this study, the primary outcomes for both arms are only 32% to 33%, then an absolute risk reduction of 35% is basically not realistic and unfounded. Therefore the study is seriously underpowered due to unrealistically estimated large effect size which led to all the non-significant results with very wide 95% confidence intervals.

2) Study design. As we are still not clear whether the convalescent plasma is useful/effective at all in treating Covid19, we would think that a RCT on plasma vs no plasma would be the priority but why on early plasma treatment vs later plasma treatment as needed? It's intuitive that if there is any effect on plasma vs no plasma, then the effect on early vs late plasma will be smaller, but an absolute risk reduction of 35%? We know that Phase II trials typically tend to find a larger effect size of over 10% but 35% seems too big and not justified.

3) Discussion. There was a light touch on the sample size in the limitation section however the small sample size issue/impact needs to be seriously and comprehensively addressed in the limitation. As this is a Phase II trial, the authors need to focus on the purpose of the phase II trials, really need to tone down the claims as seriously lacks of stats power due to small sample size, should instead focus on from the results whether a further confirmatory phase III is warranted.

4) Table 2 is a key table. Need to mention in details in the footnote a) what were adjusted in the adjusted analyses? 2) what stats models were used to give all the ORs and IRRs?

Reviewer #5: In this article, Balcells and colleagues present the results of their study titled "Early anti-SARS-CoV-2 Convalescent Plasma in Patients Admitted for COVID-19: A Randomized Phase II Clinical Trial". The study design included an early and a deferred COVID-19 convalescent plasma (CCP) treatment group. Block randomization was used to assign patients with documented SARS-CoV-2 infection who were ≤7 days from symptom onset and high risk of respiratory decompensation to the early group, which received CCP on enrollment; one unit at enrollment and one unit within 24 hours) or the deferred group, which received CCP if they met the criteria for respiratory decompensation. There were 28 patients in the early group and 30 in the deferred group, from which 13 received CCP. 

Based on their statistical analyses, the authors conclude that there was no statistical difference in composite (death, mechanical intubation, hospital stay) status on day 14 (primary outcome) or day 28 (secondary outcome). They also conclude there were no statistical differences in SARS-CoV-2 seroconversion or changes in inflammatory markers between the groups. 

Despite the negative outcome of this study, there are numerous patient characteristics and other factors that may have influenced the analysis and conclusions. The following comments and questions are provided for the authors' consideration.

1. Patients:

a. Despite randomization, the early treatment group appears to have been sicker. They had lower lymphocytes, higher ferritin and D-dimers, and more severe pneumonia on CT imaging (Table 1). In addition, more patients in this group were on steroids and fewer received anti-coagulation. Did these variables affect outcome?

b. The mean age of both cohorts was older than cohorts reported in other studies about which this reviewer is aware. Older patients may have less effective antibody (or adaptive) responses and more rapid disease progression. The range in age was wide in each group. Did outcomes vary as a function of age? 

c. Though not statistically significant, there were more deaths in the early group at 14 and 28 days (Table 2). Were the patients who died older? Sicker (lower lymphocytes, higher D-dimer, more severe pneumonia)? Did their antibody levels differ? More information on these patients should be provided. 

d. The number of patients who had SARS-CoV-2 IgG at day 3 was higher in CCP recipients, albeit not statistically significant. What were the titers? Did titers vary with age? Other patient characteristics? What were the actual titers on day 0?

2. Study design:

a. 13 of 30 patients in the deferred group received CCP. Did it affect their outcomes?

b. SARS-CoV-2 IgG positivity and neutralizing antibody levels are provided for both groups. Are IgM, IgA, or levels of antibodies to other determinants (e.g. nucleocapsid) available? 

c. Although the study was powered based on another pandemic respiratory disease, it may have been underpowered to detect differences in clinical status or mortality for COVID-19, which can be a prolonged illness.

3. Plasma:

a. Who were the CCP donors? Did all have symptomatic illness? If symptomatic, what was their degree of illness? Were they hospitalized? On what day after symptoms resolved were donations collected (the protocol says at least 28 days after symptoms)? 

b. It appears that neutralizing titers were only available on 18 of the 41 CCP units that were administered (line 288: "… all infused plasma samples tested (n=18) had a positive ID50 at screening dilution… with a median screening dilution of 97…"). What was the actual neutralizing titer of these units? 

c. The actual SARS-CoV-2 antibody titers of the CCP administered (line 124 states ≥1:400) and if available, neutralizing titers, would be helpful in assessing the possibility that CCP recipients may have received low/er titer plasma. This is underscored by the negative results of the trial in India in which CCP was deemed ineffective, but many units lacked neutralizing antibodies.

[LINK]

---

## [Decision Letter · Decision Letter 2]

16 Dec 2020

Dear Dr. Balcells,

Thank you very much for re-submitting your manuscript "Early Anti-SARS-CoV-2 Convalescent Plasma in Patients Admitted for COVID-19: A Randomized Phase II Clinical Trial" (PMEDICINE-D-20-04556R2) for review by PLOS Medicine.

I have discussed the paper with my colleagues and the academic editor and it was also seen again by four reviewers. I am pleased to say that provided the remaining editorial and production issues are dealt with we are planning to accept the paper for publication in the journal.

[LINK]

We look forward to receiving the revised manuscript by Dec 23 2020 11:59PM.   

Sincerely,

Artur Arikainen

Associate Editor

PLOS Medicine

plosmedicine.org

Requests from Editors:

1. Please address the reviewers’ final concerns, below.

2. Please amend the title to: “Early versus deferred anti-SARS-CoV-2 convalescent plasma in patients admitted for COVID-19: A randomized phase II clinical trial”

3. Abstract:

a. Line 46: Please rephrase to: “…with final follow-up until…”

b. Line 49: Please rephrase to: “…consisted of immediate…”

c. Please add a second limitation, eg. that NAbs were not confirmed in the donor plasma.

d. Line 63: Please replace “clearly show benefits” with “show effects”

4. Author Summary: 

a. Overall, we would recommend shortening your Author Summary by 1-2 bullet points per section.

b. Line 74: Please rephrase to: “The severe acute…”

c. Lines 81-82: Please rephrase to: “…few randomized clinical trials have been carried out to show any clinical benefit…”

d. Line 83: Delete this sentence: “The administration of convalescent plasma in earlier stages of COVID-19 (before respiratory failure development) was expected to be of higher benefit.”

e. Line 102: Delete this sentence: “No significant differences were observed in IgG SARS-CoV-2 seropositivity rate nor in IgG seroconversion rates between plasma receptors and no plasma receptors.”

5. Please add a space between text and the citation callouts, eg.: “…COVID-19 in the US [4].”

6. Line 185: Please remove any trademark symbols.

7. Table 1: Please remove the p value comparisons of baseline groups, as recommended by CONSORT.

8. Although you say in several places that the trial was "underpowered"; in fact you seem to have recruited the number of participants that you planned to, implying that you can indeed exclude a 35% or greater risk reduction in the primary outcome in this study (though the trial was not powered to detect a smaller reduction than that). Please clarify where appropriate.

9. Throughout, please report p values lower than 0.001 as p<0.001. Please report all other p values to 3 decimal places.

10. Throughout, please use “patients” or “participants”, instead of “subjects”.

11. Please ensure that all statements in the Abstract are also mentioned in the main text, including this sentence: “The viral clearance rate on day 3 (26% vs 8%, p=0.20) and day 7 (38% vs 19%, p=0.37) did not differ between groups.”

12. In Fig 1, it appears that 58 are shown to have completed the trial, whereas in your abstract you state 57. Please update the flowchart accordingly.

13. In the reference list, please delete publisher names (eg. “Elsevier Ltd”, “NLM (Medline)”, “Blackwell Publishing Inc.”) – the journal name is sufficient.

14. Please mark preprint references as “[Preprint]”.

15. Please provide additional details for reference 21, (eg. a URL, DOI, or ISBN).

16. When completing the CONSORT checklist, please use section and paragraph numbers, rather than page numbers.

17. Please upload a version of S2 Table without tracked changes.

Comments from Reviewers:

Reviewer #1: The paper improved but there remain some minor issues that should be resolved. 

P3 line 62. The sentence on power of the study should be correct to state something like this because it is not "possible" but it is a fact that this study was not powered to detect smaller but still potentially clinically relevant therapeutic effects of convalescent plasma. So please change to "The limitation of this study is the lack of statistical power to detect a smaller but clinically relevant therapeutic effect of convalescent plasma"

P4. Please rephrarse for clarity 

1. Our study failed to find evidence in the primary outcome with early convalescent plasma

administration compared to its use only in case of clinical deterioration

Should be: Our study failed to show that early convalescent plasma administration improves the outcome compared to its use only in case of clinical deterioration

2. This evidence could be limited by the small sample size of the study.

Should be: The small sample size of the study precludes any definite conclusions but the results are in agreement with observations from other trials on convalescent plasma for patients hospitalized with COVID19 (ref. recent BMJ paper on study in India, Medriv paper on Dutch study from Gharbharan et al. etc )

P5 line 128: please change to something like this : but its effect on overall mortality remains controversial (ref Solidatory trial and ref final report of ACTT study)

P6 line 157: please change "but" to and imaging ….

P7 line 179: should be anti-HLA antibodies

P8 line 189-94: please move this part to results section

P8 line 193 : were these 18 plasma units (not plasmata) from 18 different donors? If yes please change to "In 18 of the 40 plasma donors virus neutralizing antibody titers were measured". If no, then the number of donors not plasma units that was tested should be mentioned instead

P11b line 257. Please remove "of"

P12 line 272: Only a p-value for the primary endpoint can be considered statistically significant at the 0,05 cut-off. So please add : For the primary endpoint … and add "The statistical analysis of secondary endpoints should be considered exploratory only"

Line 324: was higher (not resulted higher)

Line325: remove "develop"

Discussion;

Line 456: I don't think it was approved. It received an emergency use authorization. 

Line 468: Secondly, the study was probably underpowered to detect a statistically significant difference and the wide confidence intervals that we found for the effects on main outcomes do not permit to exclude that convalescent plasma may show clinical benefit in a future larger clinical trial.

Please change to: Secondly, the study was only powered to detect a huge impact on the primary endpoint under study as illustrated by the wide confidence interval. Larger trials are therefore required to exclude smaller but clinical relevant effects. 

474: on early stage => in an early stage …

Reviewer #3: Thank you for addressing my concerns as well as the concerns of the other reviewers in your revision.

Reviewer #4: Thanks authors for their great effort to improve the manuscript. The authors have addressed my concerns/comments adequately. I am satisfied with the response and revision. No further issues needing attention.

Reviewer #5: In this revised article, Balcells and colleagues have answered the major concerns of the prior review pertaining to study design and sample size calculation. The main result of the study is that there was no difference in clinical outcome of hospitalized patients with Covid-19 randomized to early versus delayed treatment with convalescent plasma. The baseline clinical data of the cohort suggest that the patients had severe disease and were at high risk for clinical deterioration, despite being enrolled ≤ 7 days after symptom onset. Neutralizing antibody titers were measured on 18 of 41 units of convalescent plasma that were administered. There are multiple factors that may have precluded a beneficial effect of plasma in this study, including that the patients had pneumonia and evidence of inflammatory progression at enrollment, were at risk for deterioration based on disease severity and age, and the administered plasma may not have had sufficient antibody to mediate an antiviral effect. These points, which are addressed well by the authors, are important to the field as it wrestles with the challenge of establishing a possible benefit of convalescent plasma and informing clinical guidance for its use. There are a few points that should be addressed to increase clarity. 

Lines 420-422: The statement that "…convalescent plasma was not associated with a higher rate of SARS-CoV-2 clearance … suggesting it does not provide a strong antiviral activity at this stage" may be true. However, the data do not support this conclusion - neutralizing titers were determined on less than half of the units administered and some of the units had neutralizing titers that were < 1:160. This statement should be modified. In addition, it would be helpful to present the actual Ct values rather than the ratios. It is not possible to determine the patients' viral burdens from the ratios. 

Lines 423-426: The lack of antibody positivity in transfused patients is curious. Does the group of patients who did not exhibit a rise in antibody level include patients who did not have antibody at enrollment? The higher neutralizing titers of patients with a longer symptom duration is consistent with other studies (see Garcia-Beltran et al. medRxiv).

[LINK]

---

## [Editor Report · Decision Letter 3]

6 Jan 2021

Dear Dr. Balcells,

Thank you very much for re-submitting your manuscript "Early versus Deferred Anti-SARS-CoV-2 Convalescent Plasma in Patients Admitted for COVID-19: A Randomized Phase II Clinical Trial" (PMEDICINE-D-20-04556R3) for review by PLOS Medicine.

I am pleased to say that provided the remaining editorial and production issues are dealt with we are planning to accept the paper for publication in the journal.

The remaining minor issues that need to be addressed are listed at the end of this email. Any accompanying reviewer attachments can be seen via the link below. Please take these into account before resubmitting your manuscript:

[LINK]

We look forward to receiving the revised manuscript by Jan 13 2021 11:59PM.   

Sincerely,

Artur Arikainen, 

Associate Editor 

PLOS Medicine

plosmedicine.org

Requests from Editors:

1. Line 234: Papers cannot be cited until they have been accepted for publication or are publicly available on a preprint archive. The information may be cited in the text as a personal communication with the author if the author provides written permission to be named. Alternatively, please provide a different appropriate reference. 

2. Please delete “[Internet]” from references of journal articles: 32, 33, 35.

---

## [Editor Report · Decision Letter 4]

12 Jan 2021

Dear Dr Balcells, 

On behalf of my colleagues and the Academic Editor, Sanjay Basu, I am pleased to inform you that we have agreed to publish your manuscript "Early versus Deferred Anti-SARS-CoV-2 Convalescent Plasma in Patients Admitted for COVID-19: A Randomized Phase II Clinical Trial" (PMEDICINE-D-20-04556R4) in PLOS Medicine.

PRESS

Sincerely, 

Artur A. Arikainen 

Associate Editor 

PLOS Medicine